# Emergent Nanotechnological Strategies for Systemic Chemotherapy against Melanoma

**DOI:** 10.3390/nano9101455

**Published:** 2019-10-13

**Authors:** Jacinta Oliveira Pinho, Mariana Matias, Maria Manuela Gaspar

**Affiliations:** Research Institute for Medicines, iMed.ULisboa, Faculty of Pharmacy, Universidade de Lisboa, Av. Prof. Gama Pinto, 1649-003 Lisboa, Portugal; jopinho@ff.ulisboa.pt (J.O.P.); mariana.r.matias@gmail.com (M.M.)

**Keywords:** melanoma, nanotechnology, lipid-based nanosystems, systemic chemotherapy, therapeutic targets

## Abstract

Melanoma is an aggressive form of skin cancer, being one of the deadliest cancers in the world. The current treatment options involve surgery, radiotherapy, targeted therapy, immunotherapy and the use of chemotherapeutic agents. Although the last approach is the most used, the high toxicity and the lack of efficacy in advanced stages of the disease have demanded the search for novel bioactive molecules and/or efficient drug delivery systems. The current review aims to discuss the most recent advances on the elucidation of potential targets for melanoma treatment, such as aquaporin-3 and tyrosinase. In addition, the role of nanotechnology as a valuable strategy to effectively deliver selective drugs is emphasized, either incorporating/encapsulating synthetic molecules or natural-derived compounds in lipid-based nanosystems such as liposomes. Nanoformulated compounds have been explored for their improved anticancer activity against melanoma and promising results have been obtained. Indeed, they displayed improved physicochemical properties and higher accumulation in tumoral tissues, which potentiated the efficacy of the compounds in pre-clinical experiments. Overall, these experiments opened new doors for the discovery and development of more effective drug formulations for melanoma treatment.

## 1. Introduction

Cancer incidence and mortality are rapidly increasing worldwide, with an estimated 18.1 million new cancer cases and 9.6 million cancer deaths in 2018. Moreover, according to the World Health Organization (WHO) in 2015, cancer was the first or second leading cause of death before the age of 70 years in 91 countries and the third or fourth cause of death in an additional 22 countries [1]. These values reflect the extension of cancer that persists as one of the major global public health concerns.

Chemotherapy, as a monotherapy or drug combination, is the most used therapeutic approach for the treatment of cancer, increasing the number of long-term cancer survivors. However, anticancer agents have several and significant limitations that frequently compromise the effectiveness of the therapy and the continuation of the treatment [2]. The standard chemotherapeutic agents preferentially act on dividing cells by inducing DNA damage and strand breakage, which interferes with DNA repair and microtubule function, specifically vinca alkaloids and taxanes [3,4]. These mechanisms are nonspecific and can result in damage to healthy tissues in addition to tumor cells [3]. Another relevant and alarming obstacle in the treatment of cancer is the development of drug resistance by tumor cells. This pharmacoresistance can be due to intrinsic factors, which include mutations, gene amplifications, deletions and chromosomal rearrangements or extrinsic factors, such as pH, hypoxia and paracrine signaling interactions with stromal and other tumor cells [5]. One possible strategy to help overcome multi-drug resistance is the delivery of combination chemotherapy that acts on different targets and, consequently, affects several signaling pathways [6]. 

Taking into account these issues, in this review, we endeavor to summarize the investigation of novel small molecules together with the identification of new therapeutic targets against melanoma. Moreover, the application of nanotechnology to solve this problem has been an intelligent and fruitful strategy to improve selectivity for diseased tissues, enhance the efficacy and biochemical properties of active molecules and reduce pharmacoresistance [7,8,9]. Thus, the exploitation of new potential drugs for melanoma treatment is herein addressed, with the focus on lipid-based nanoformulations (e.g., liposomes) as efficient drug delivery systems.

To prepare this review, an extensive literature search from two databases—PubMed and Science Direct—was performed to generate a critical and comprehensive overview of potential targets for melanoma treatment and the most recent advances in the search for new or more potent molecules formulated in lipid-based nanosystems, with anticancer activity against melanoma. The keywords for the search consisted of combinations of the following terms—melanoma, aquaporin-3, tyrosinase, lipid-based nanosystems and liposomes.

## 2. Melanoma—General Overview

Melanoma arises from the malignant transformation of melanocytes, cells responsible for melanin production. Several factors contribute for the onset of the disease, namely sun exposure and other environmental factors, genetic predisposition and immunosuppressive states [10]. Melanoma is considered the most aggressive and deadly form of skin cancer, with increasing incidence and mortality worldwide [11,12]. The adopted treatment is chosen according to the stage of the disease. When detected and treated at an early stage, most melanoma cases are curable by surgery. However, once it progresses to the metastatic state, treatments often fail, leading to 80% of deaths related to skin cancers [13]. Melanoma cells are able to metastasize in nearby tissues and in distant major organs, with the most common metastatic sites being located in the lungs, brain, lymph nodes, liver and bone [14]. It is of note that, among all human cancers, melanoma presents the highest mutation burden. This biological heterogeneity allows melanoma cells to resist and evade the adopted therapies [15,16]. As a result, despite the substantial advances on cancer clinical management, treatments against metastatic melanoma remain non-effective or relapses are often observed [17,18].

### 2.1. Current Therapeutic Approaches

The constant technological evolution and the increasing understanding of cell and tumor biology boosted the advances on cancer therapeutic options (Table 1). Nevertheless, the results against metastatic melanoma remain disappointing, as treatments are, in most cases, non-effective or relapses are observed. Also, drug resistance is a prominent feature of melanoma, which leads to the lack effectiveness of current therapies and the ability of tumors cells to develop mechanisms enabling their survival within the host [17,19,20]. Next, melanoma therapeutic approaches are briefly described.

#### 2.1.1. Surgery

When the disease is detected early (stages I and II), the surgical removal of melanoma can be successfully achieved, with relatively low morbidity [21]. This medical procedure may prevent the occurrence of metastasis; however, in most cases of advanced melanoma, the cancer cannot be eradicated through this approach. Notwithstanding, clinical trials combining surgical resection with systemic therapies have been conducted in melanoma at stages III and IV [21,22].

#### 2.1.2. Chemotherapy

Some decades ago, dacarbazine (DTIC; 1970s) [23], an alkylating agent with cytostatic activity, was the first chemotherapeutic drug clinically approved for melanoma. Temozolomide [24], an analog of DTIC and approved for glioblastoma multiforme, is often used in patients with advanced melanoma since it is orally administered, penetrates the central nervous system and displays a favorable toxicity profile [25,26,27]. However, these compounds do not show significant therapeutic benefits and lead to adverse events, since they lack specificity towards tumor sites and melanoma cells often developed resistance to alkylating agents [25,26,27].

#### 2.1.3. Radiotherapy

Cancer cells subjected to radiotherapy suffer DNA damage. At low dose, melanoma cells are able to effectively repair these damages due to high proliferation capacity, efficient enzymatic system and poor cell differentiation [28,29]. Therefore, this treatment modality has a limited application for melanoma patients, being selected for specific clinical cases. Radiotherapy alone is considered when the wide excision of primary tumor is impractical [28]. Moreover, in case of high recurrence risk after surgery, radiotherapy is applied as an adjuvant approach, improving local control of the primary tumor site [28,30,31]. Radiotherapy is still an option in melanoma management, with advances being accomplished in terms of real-time imaging and improved dose regimen according to disease progress. In addition, the combination of radiotherapy with existing systemic therapeutic approaches is under study [29].

#### 2.1.4. Targeted Therapy

With the evolution of the pharmaceutical field, targeted therapies became a promising field of research. For instance, the mitogen activated protein kinase/extracellular signal-related kinase (MAPK/ERK) signaling pathway regulates cell growth, proliferation, differentiation, migration and survival of all mammalian cells [32,33]. Approximately 50% of melanoma cases bear the BRAF V600E somatic mutation, leading to dysregulation of the MAPK signaling pathway and promoting tumor cell growth. This knowledge prompted the design, development and clinical approval of selective small molecule inhibitors of serine/threonine protein kinase B-raf (BRAF) and mitogen activated protein kinase (MEK) [16,33,34,35]. Nevertheless, despite the initial promising results, when not used in combination, these agents are unable to improve survival and tumor resistance to therapy and relapses are frequently observed [36,37].

#### 2.1.5. Immunotherapy

Immunotherapy is another expanding area of melanoma treatment research. In its initial stages, melanoma is known to be the most immunogenic type of cancer [38,39], although this feature is lost when melanoma reaches the metastatic state, manipulating the microenvironment and abolishing the immune responses [40]. Between 1985 and 1993, research on immunotherapy with interleukin-2 (IL-2), a cytokine that promotes T cell growth, greatly progressed, with its approval by the Food and Drug Administration (FDA), in 1998, as the first immunotherapy for advanced melanoma [32,39]. More recently, the blockade of cytotoxic T-lymphocyte-associated protein-4 (CTLA-4) and programmed cell death protein 1 (PD-1) have enhanced T-cell mediated antitumor immunity [33,39,41,42]. Although the clinical data are promising, low percentages of effective and prolonged responses, resistance or relapse and adverse effects have been observed for immune checkpoint therapy [39,43,44,45]. To circumvent these difficulties and improve clinical outcomes, innovative research on delivery platforms for this type of therapy is starting to emerge [46].

In order to overcome the critical limitations of drug resistance and lack of specificity of therapies, continuous research efforts are being employed to develop more effective therapeutic options. In this sense, the identification of novel therapeutic targets, as well as the development of compounds with potential anticancer activity, are being accomplished [20].

## 3. Advancing Melanoma Systemic Treatment—Potential Targets and Therapeutic Agents

The development of successful drug candidates begins with the understanding of the disease and the identification of putative therapeutic targets, leading to optimized drug-target interactions. Several targets have been associated to melanoma pathogenesis, such as tyrosinase [47,48,49], aquaporin-3 (AQP3) [50,51], folate receptor (FR) [52], integrin αvβ3 [53], cyclooxygenase-2 [54], STAT3 [54], protein phosphatase 1 [55], cytosolic phospholipase A_2_ [56], melanocotin-1 receptor [57], topoisomerase 1 [58], prolyl isomerase Pin1 [59] and actin microfilaments [60]. The use of nanotechnological tools for the modulation of these therapeutic targets will be addressed in this manuscript. In addition, in the context of therapeutic targets, an emphasis will be given in the following sections to AQP3 and tyrosinase, which are up-regulated in melanoma.

### 3.1. Aquaporins—Structure and Function

All biological membranes display an intrinsic water permeability that, depending on membrane lipid composition, allows cell volume to equilibrate in minutes or less in response to an osmotic gradient [61,62]. Nevertheless, in tissues that require a higher water permeability, such as fluid secretion and absorption, cells are able to regulate water transfer more efficiently through aquaporins (AQPs), membrane proteins that are ubiquitous in all domains of life [61]. In 1992, aquaporin-1 (AQP1) was the first protein identified as water channel [63]. Aquaporins belong to a family of 13 small (molecular size of ~30 kDa) transmembrane channel proteins that are ubiquitous to all living organisms, which emphasizes their crucial role in maintaining fluid homeostasis [64,65,66]. Individual monomers consist of six membrane-spanning helical domains and two short helical segments that surround the intra—and extracellular vestibules connected by a narrow aqueous pore. Monomers assemble to form tetramers in the plasma membrane, with each monomer functioning independently as a channel [61,66].

Aquaporin channels can be grouped into two main categories: orthodox (AQPs 0, 1, 2, 4, 5, 6 and 8), which are water-specific channels and aquaglyceroporins (AQPs 3, 7, 9, 10), involved in the bidirectional transport of water and small polar solutes, namely glycerol and urea [66]. Also, peroxiporins comprise a subclass of AQPs that are permeable to hydrogen peroxide (H_2_O_2_) and include AQP1, AQP3, AQP5, AQP8 and AQP9. Finally, AQP11 and AQP12 display a subcellular localization and, as such, are classified as S-aquaporins [67,68,69]. These last two proteins remain poorly understood and, to some extent, their structure and subcellular distribution distinguishes them from the other groups [70].

#### Aquaporin-3 as Potential Therapeutic Target in Melanoma

As crucial players in normal human physiology, AQPs regulate numerous physiological cell functions including energy metabolism, protein expression, cell volume, adhesion, proliferation, differentiation and migration, as well as apoptosis [66,71,72]. Evidence regarding the roles of several AQPs in cancer development and progression are emerging, with different AQPs being associated to several tumor types and regulating cells proliferation, migration and angiogenesis [69,73].

As stated above, AQP3 is permeable to water, glycerol [74], ammonia [66], urea [66,74] and H_2_O_2_ [75]. This aquaglyceroporin is known to be expressed in different tissues, including skin [76], respiratory tract [77], kidneys [74] and gut [78]. AQP3 has also been found to be present in distinct cancer types [79,80,81]. In melanoma, it is well-known that AQP3 is overexpressed [50,51] and researchers have shown that AQP3 knockout mice are resistant to skin tumor formation [79]. A possible mechanism for this impaired skin tumorigenesis is the AQP3-mediated H_2_O_2_ transport in AQP3 null mice [82]. Also, Gao and coworkers [50] reported that promoting the overexpression of AQP3 in human melanoma cells would increase chemoresistance to arsenite [50]. Arsenic-based compounds have shown potential as chemotherapeutic agents, with the FDA approval, in 2001, of arsenic trioxide (Trisenox^®^) for the treatment of acute promyelocytic leukemia [83].

Considering their participation in normal and diseased physiological states, the pharmacological modulation of AQPs emerges as a promising opportunity for the development of novel and innovative therapeutic strategies in a variety of human disorders, namely cancer. Despite being promising therapeutic targets, the identification of potential AQPs modulators has proved to be a demanding task [66,84,85]. Until now, four classes of AQP-targeting agents have been defined—(1) metal-based inhibitors (Figure 1); (2) small molecules described as being water conductance inhibitors; (3) small molecules targeting the interaction between AQP4 and the neuromyelitis optica autoantibody; and (4) agents that act as chemical chaperones, causing AQP2 mutants [66].

### 3.2. Therapeutic Potential of Metal-Based Compounds

Metallodrugs have proven to hold great potential as therapeutic agents in several diseases [86,87]. The discovery [88] and clinical approval of the anticancer drug cisplatin [cis-diamminedichlorido platinum(II)] was a remarkable achievement that prompted the research on metal-based complexes as biologically active agents [89]. Currently, three platinum-based drugs, cisplatin, carboplatin and oxaliplatin, are in clinical use for cancer treatment. Additionally, ruthenium (Ru) complexes have emerged as promising second-generation metal-based anticancer agents. Moreover, some of them have entered in clinical trials [90]. Particularly, Ru(III)-containing nucleolipids showed a remarkable in vitro anticancer activity [90]. More recently, Ru(III)-complexes incorporated into a DOTAP liposomal formulation demonstrated effective antitumor activity both in vitro and in vivo [91]. Over the years, several metal-based compounds for the treatment of different human pathologies have been clinically approved, namely for cancer (platinum and iron), microbial infections (silver), arthritis (gold), ulcers (bismuth), protozoan infections (antimony), diabetes (vanadium) and malaria (iron) [66,92,93]. In contrast to organic compounds, the metal-based ones have a wide range of coordination numbers and geometries, as well as kinetic properties, offering novel mechanisms of drug action that, otherwise, would be unavailable [94].

Following this widespread success, the field of coordination chemistry has greatly progressed towards the development of improved metallodrugs, including gold—and copper-based complexes [66,87,95,96,97,98,99,100,101]. Compounds of copper (mostly Cu^2+^) are being studied as new generation metallodrugs based on the notion that endogenous metals may be less toxic for normal cells, opposed to cancer cells [102,103]. Copper, a biologically active metal ion, possesses distinctive hydrolytic and redox properties. The Cu^2+^ is able to form complexes with various coordination numbers and geometries, offering promising therapeutic applications [104,105]. As an example, the cytotoxic binuclear Cu^2+^ complex [Cu(phen)_2_]^+^ (phen = 1,10-phenanthroline) has been reported to bind DNA and to induce single-stranded breaks [106]. Moreover, copper complexes with a phenanthroline-type ligand, from the *Casiopeína* class and with the general formula [Cu(N–N)(O–N)]^+^ or [Cu(N–N)(O–O)]^+^, have entered clinical trials [107]. For instance, the Cu^2+^ complex named Casiopeína IIgly is being studied as a potential new anticancer drug. It induces reactive oxygen species (ROS)-mediated mitochondrial dysfunction, ultimately resulting in apoptosis [108,109]. Also, Slator and coworkers [110] investigated the Cu^2+^ complex [Cu(o-phthalate)(phenanthroline)] as an intracellular ROS-active cytotoxic agent [110]. An interesting approach of Jaividhya and colleagues (2015) was the development and study of fluorescent mixed ligand copper(II) complexes. These could prove to be advantageous for detecting the compounds within the target sites, facilitating the understanding of their interaction with cells [111].

In addition to copper, gold has also been explored as an anticancer agent since it was found that patients with rheumatoid arthritis receiving gold(I)-based drugs were less prone to cancer development [112]. Also, these gold(I) compounds were subsequently found to inhibit the growth of HeLa cells [112]. In relation to gold(III) complexes, the design of new ligands has improved the stability of the complexes in the reducing milieu of biological systems. This has prompted the research on the use of gold(III) compounds as potential antitumor agents [113,114].

#### Metal-Based Compounds as AQPs Inhibitors

Mercurial compounds were the first to be described as water permeability blockers through AQPs [115,116]. Other heavy metals, such as silver [117] and zinc [118], have also been and continue to be explored.

Researchers have demonstrated that copper(II) ions inhibit AQP3 [119,120], reducing cell growth and increasing cisplatin’s therapeutic effects [120]. In addition, copper(II) was shown to inhibit AQP3 in a rapid and reversible way and that this effect did not require its internalization by cells. The same authors suggested that the copper-mediated AQP3 inhibition involves three amino acid residues located in the extracellular loops (Trp128, Ser152 and His241) [119]. Moreover, selective and potent gold (Auphen; [Au(phen)Cl_2_]Cl) [95,96,121,122] and copper-based (Cuphen; [Cu(phen)Cl_2_]) [97,101] (Figure 2) inhibitors of AQP3 have recently been reported as advantageous to target tumor cells overexpressing this aquaglyceroporin. For instance, researchers evaluated the antiproliferative effects of Auphen on several tumor cell lines with different AQP3 expression levels—no expression (PC12 cells; rat adrenal gland pheochromocytoma), high expression (A431; human epidermoid carcinoma cells) and overexpression (PC12-AQP3; PC12 cells transfected with AQP3). The authors demonstrated that Auphen antiproliferative activity was positively correlated with AQP3 expression by specifically affecting AQP3-mediated glycerol permeability [96,121].

Despite the constant evolution of the coordination chemistry field, the progress of promising metal-based complexes in clinical trials is often hindered by inherent toxic side effects [123] and ‘speciation’ [124]. To overcome these current limitations, new strategies should be adopted, such as the use of nanotechnological tools for targeted delivery.

### 3.3. Tyrosinase—General Overview

Pigmentation is a process limited to melanocytes and the retinal pigment epithelium. Human tyrosinase, a melanosomal glycoprotein, is involved in the initial steps of melanin pigment biosynthesis, catalyzing the hydroxidation of l-tyrosine to l-3,4-dihydroxyphenylalanine (l-DOPA) and oxidation of l-DOPA to dopaquinone [47,125,126]. Tyrosine structure consists of four conserved regions, namely a small C-terminal cytoplasmic domain, a single transmembrane α-helix, a N-terminal signal peptide and an intra-melanosomal domain (catalytic domain). The catalytically active core of the enzyme contains two copper centers (Cu (II) A and Cu (II) B) close to each other (*d*_Cu-Cu_ from 2.9 to 4.9 Å) and primarily coordinated to three histidine residues [47,48,49].

A well-established and effective approach to regulate the production of melanin in vivo is the inhibition of tyrosinase. Consequently, the development of tyrosinase inhibitors has greatly impacted medicine and cosmetics (whitening agents), as well as agricultural industry (insecticides and browning inhibitors for vegetables/fruits) [48,49,127]. An immense diversity of compounds has been described and employed as tyrosinase inhibitors, including those of natural-based and synthetic origin [127,128].

#### 3.3.1. Tyrosinase as Potential Therapeutic Target in Melanoma

Several years ago, researchers began studying a putative connection between melanin biosynthesis and melanoma progression and resistance, as well as the modulation of melanogenesis as antimelanoma therapy [49,129]. Opposed to normal melanocytes, melanoma cells are able to retain the synthesized melanin, instead of releasing it to adjacent keratinocytes. In turn, this excess of pigment (a) protects melanoma cells from radiotherapy; (b) produces melanogenesis intermediates that are selectively toxic towards immune cells; and (c) functions as an antioxidant, counteracting chemotherapy effects [49,130].

Taking into account all these facts, the potential of tyrosinase as a therapeutic target is undeniable, as this enzyme is overexpressed in melanoma and associated with tumor development and progression [47]. This augmented abundance of tyrosinase can be explored to selectively direct therapeutic compounds at melanoma sites, a strategy termed as “Melanocyte-Directed Enzyme Prodrug Therapy” (MDEPT, Figure 3) [131]. This tyrosinase-mediated drug delivery consists of the recognition and oxidation of the prodrugs pro-moiety and subsequent release of the cytotoxic agent [49,132,133]. Improved drug physicochemical and pharmacokinetic properties are frequently attained by the use of these prodrugs, defined as derivatives/precursors of therapeutically active molecules. These may be converted in the active drug by enzymatic and/or chemical processes in vivo [134].

#### 3.3.2. Tyrosinase-Activated Prodrugs

In the first attempts of MDEPT, the selected cytotoxic agent was attached to existing tyrosinase substrates, namely tyrosine and dopamine, by means of carbamate or urea linkers (Figure 4a,b) [135,136,137]. Unfortunately, these initial studies failed, as the release of the active compound did not occur [138]. In the following years, researchers focused on tyramine and dopamine derivatives of triazenes as potential tyrosinase-activated prodrugs for melanoma treatment [132]. Triazenes belong to a well-known class of anticancer drugs used against melanoma, which have a *N*=*N*–*N* structure type, normally next to an aromatic ring. Currently, DTIC and temozolomide are the only triazenes with clinical application [133]. The synthetized triazene-derived compounds by Perry and collaborators [132] were highly stable in human plasma and good tyrosinase substrates. However, the release of the cytotoxic agent was not observed for the urea-linked derivatives. The authors concluded that future synthetized molecules should have improved tyrosinase-mediated drug release, while maintaining their stability in physiological milieu [132].

In more recent years, a novel prodrug containing a hydrazine linker was synthesized [139]. However, these phenylhydrazine derivatives were quickly abandoned as an option due to their intrinsic toxicity and lack of tyrosinase selectivity, displaying higher susceptibility to oxidation by different enzymes (Figure 4c) [139].

Monteiro and coworkers [133] have designed and developed prodrugs with high cytotoxicity against different human melanoma cell lines (SKMEL-30, MNT-1 and M8). In addition, a correlation between higher cytotoxicity and higher tyrosinase activity was achieved, indicating a tyrosinase-dependent activation mechanism of the prodrugs [133]. In the latest work of Sousa and colleagues (2017) [140], a new series of triazene derivatives (triazene hybrid molecules) with different physicochemical properties were synthetized. These compounds were chemically stable in physiologic milieu and, when in contact with tyrosinase, the alkylating moiety was released. The most promising compounds displayed high affinity towards tyrosinase and, consequently, high cytotoxicity against MNT-1 and B16F10 cell lines, which overexpress tyrosinase. Remarkably, two of these derivatives were demonstrated to be more cytotoxic than temozolomide. Moreover, low toxicity towards healthy human keratinocytes (HaCaT) was observed, reinforcing the strong potential of this strategy for melanoma treatment [140].

## 4. Nanotechnology and Cancer Treatment—Tackling Melanoma

In order to overcome the current limitations of melanoma treatment, considerable endeavors have been made to promote selective delivery to diseased tissues. The first steps in the biomedical application of nanotechnology began more than 50 years ago, when the first artificial lipidic membranes were developed [141] and their usefulness as delivery tools for pharmacological active substances began to be unveiled [142]. Since then, nanotechnology has been recognized as a paradigm-changing factor in cancer management, showing a tremendous progress through time. 

The effective delivery of treatment to melanoma tumor sites is a very complex and challenging goal due to the particular features of solid tumor biology. In this context, nanosystems offer the opportunity to combine both passive and active tumor targeting.

In a solid tumor setting, the vasculature has an abnormal structure and impaired function, leading to a deficient blood flow, increased hypoxia and higher interstitial fluid pressure (IFP) [143,144,145]. For passive targeting, two major aspects have been researched and explored: the leaky nature of tumor blood vessels and the deficient lymphatic drainage. This allows the extravasation and preferential accumulation of nanosystems at solid tumor sites owing to the Enhanced Permeation and Retention (EPR) effect (Figure 5) [146,147,148,149]. This characteristic is globally recognized as an advantageous opportunity to promote passive tumor accumulation of nanoparticles and to facilitate their cellular uptake, depending on the nanoparticles physicochemical properties [148,149]. Moreover, compared to healthy tissues, tumor microenvironment has unique physicochemical features, including hypoxia, slightly acidic pH, active efflux pumps, hyperthermia and overexpression of several molecular biomarkers [150,151,152]. Considering this, a locally triggered drug release from nanosystems may be promoted by taking advantage of these intrinsic tumor biological peculiar conditions (e.g., pH, temperature and enzymes) or through the application of external stimuli (e.g., light, electric fields, magnetic fields or ultrasound) [144,152,153,154,155,156,157].

In the case of active targeting, ligands (whole antibodies or their fragments, peptides, aptamers, small molecules, among others) are bound to the nanoparticles’ surface and used as an advantageous modification to achieve a selective targeting towards tumor cells [159].

Among the several drug delivery tools, lipid-based nanosystems have been successfully used to incorporate various compounds, demonstrating suitable biological properties, including biocompatibility, biodegradability and the ability to accommodate both hydrophilic and hydrophobic active molecules [160].

To surpass the inefficacy and toxicity of conventional available drugs in melanoma treatment, several types of nanosystems have been explored for their applications in targeting this aggressive disease. In light of the continuous progresses, this review addresses the development and study of lipid-based nanosystems as efficient delivery vehicles of new bioactive molecules for targeting melanoma, focusing on liposomal formulations.

### 4.1. Liposomes

Among the diversity of lipid-based nanosystems currently available, liposomes (Figure 6) are undoubtedly the most well-known and versatile due to their unique properties. Liposomes are composed of one or more concentric lipid bilayers separated by aqueous compartments. Their major component, phospholipids, can be combined with other molecules, including glycolipids, cholesterol (Chol) and other amphipathic substances [160,161,162]. Liposomes as drug delivery systems have been extensively studied and, based on their ability to incorporate both hydrophilic or hydrophobic molecules, preclinical studies demonstrated an improved therapeutic performance of associated compounds. Many successful applications are already in clinical use (Table 2) or under clinical trials [163,164,165].

Liposomes provide a repertoire of therapeutic advantages when used as a drug delivery tool, including: (1) biocompatibility and low immunogenicity; (2) may improve the solubility and stability of drugs; (3) avoid premature drug degradation, increasing the half-life; (4) allow for a preferential accumulation at tumor sites, increasing local drug concentration; (5) may be sterically stabilized by associating hydrophilic molecules onto their surface, namely polyethylene glycol (PEG); (6) may be specifically targeted by surface modification with specific ligands [160,166,167,168,169,170,171].

According to the desired therapeutic goal, several parameters should be taken into account when designing liposomal formulations, namely by lipid composition, structure, mean size and superficial charge [164,167]. For instance, the size of liposomes greatly influences loading capacity, as well as the stability and biodistribution profile of the associated molecules. On the one hand, sizes between 80 and 200 nm provide increased stability, as well as improved extravasation to tumor sites. On the other hand, larger liposomes possess a higher loading efficiency; however, these are less stable and their clearance from blood is faster [160,172].

Moreover, the slightly acidic tumor microenvironment can be explored as a stimulus to promote local drug release from liposomes. For this, pH-sensitive properties take advantage of the polymorphic phase behavior of unsaturated phosphatidyl ethanolamine (PE), such as dioleoyl phosphatidyl ethanolamine (DOPE) that, due to its unsaturated chains, forms an inverted hexagonal phase, rather than bilayers [173,174,175]. Liposomes containing DOPE in the lipid composition may be stabilized into bilayers by using an acid lipid, such as oleic acid (OA), linoleic acid (LA) and cholesteryl hemisuccinate (CHEMS), which are negatively charged at a neutral pH. In the case of CHEMS, its homogenous distribution decreases DOPE intermolecular interactions, preventing hexagonal phase formation under physiological conditions. At a lower pH, CHEMS molecules change their structure, destabilizing the lipid bilayer and promoting the release of incorporated drugs [173,174,175].

The clinical acceptance of liposomes as drug carriers has been reflected in a number of liposome-based formulations already approved (Table 2). With respect to the use of liposomes as drug delivery platforms for melanoma treatment, encouraging in vitro and in vivo data are available in the literature [176,177].

### 4.2. Lipid-Based Nanosystems for the Delivery of New Bioactive Molecules

As aforementioned, there is an urgent need for the development of more effective and safe therapeutic options for melanoma. One strategy to overcome the current drawbacks is the incorporation or encapsulation of biologically active agents in lipid-based systems (Figure 7). Thus, in this section, the available scientific evidence on the antitumoral activity of investigational nanoformulated compounds against melanoma is critically discussed. In this context, the best nanoformulation(s) for each compound are described in Table 3 and a summary of the main results of in vitro and in vivo preclinical studies that support this activity is presented in Table 4 and Table 5, respectively.

#### 4.2.1. Liposomal Formulations of Synthetic Compounds

As referred to above, Cuphen is a potent inhibitor of AQP3 and is being explored as an anticancer drug candidate. This compound was incorporated in liposomes (Table 3 C1) and showed a notable in vitro antiproliferative activity (Table 4, F1) against several human tumor cell lines. Moreover, in vivo studies demonstrated no hepatic toxic side effects after parenteral administration of these liposomes in healthy mice [97]. Following these promising data, this copper-based compound was efficiently loaded into optimized CHEMS-containing liposomes, which promoted a pH-dependent Cuphen release (Table 3, C2). In a murine melanoma model, the treatment with this nanoformulation significantly impaired melanoma progression in vivo, devoid of hepatic toxic side effects, rendering it an attractive approach (Table 5, F1) [101].

Additionally, a new naphthalenediimide derivative (AN169, Figure 8) with promising anticancer activity was incorporated into PEGylated liposomes by Parise and collaborators [178]. This compound was successfully loaded into liposomes with a high entrapment efficiency (Table 3, C3) and the in vitro antitumor activity against human melanoma cells was evaluated. In these studies, a preservation of the cytotoxic properties was observed, as demonstrated by similar IC_50_ values, when compared to the free compound (Table 4, F3) [178].

In another study [179], an indole derivative, methyl 6-methoxy-3-(4-methoxyphenyl)-1*H*-indole-2-carboxylate (MMI) (Figure 8) was evaluated in vitro as a potential inhibitor of human melanoma cells (A375-C5) growth, exhibiting a quite low GI_50_ (50% of cell growth inhibition) value of 0.33 μM. Afterwards, this potential antitumor compound was encapsulated in different nanosized liposomes, the two most promising ones are detailed in Table 3 (C4). Both nanoformulations were monodisperse and stable after 2 weeks, with no evidence of aggregation, holding potential for future applications as an antitumor strategy [179]. The same research group also successfully incorporated a synthetic hydrophobic compound, methyl 3-amino-6-(benzo[*d*]thiazol-2-ylamino)thieno[3,2-*b*]pyridine-2-carboxylate, (MATP, Figure 8) in liposomes (Table 3, C5). This compound displayed a high affinity towards serum albumin, enabling its transportation in the bloodstream. In contrast, MATP showed low binding affinity to human multidrug resistance protein MDR1, a membrane drug efflux pump that promotes pharmacoresistance and the ineffectiveness of drug cancer therapy. This was an indication that tumor cells may not be able to expel this synthetic compound through this mechanism. Overall, the authors emphasize the potential of liposomal MATP for antitumor applications [180].

Since tyrosinase is a putative therapeutic target against melanoma, a sulfur homolog of tyrosine, 4-*S*-cysteaminylphenol (4-*S*-CAP, Figure 9) was synthetized [181]. Further, it was shown to be a suitable tyrosinase substrate, leading to selective cytotoxic effects against melanocytes and melanoma cells [181]. Subsequently, in order to combine chemotherapy and hyperthermia modalities against malignant melanoma, 4-*S*-CAP was incorporated in magnetite cationic liposomes (Table 3, C6) [182]. The nanoformulation exerted cytotoxic effects on B16 melanoma cells (Table 4, F4), being less toxic to normal human dermal fibroblasts (NHDF). Moreover, when cells were treated with nanoformulated 4-*S*-CAP combined with hyperthermia, an additive therapeutic effect was achieved. In a B16 murine melanoma model, in mice receiving liposomal 4-*S*-CAP and hyperthermia, tumor growth was strongly suppressed, with complete regression of 17% subcutaneous melanoma tumors [182].

Taking into consideration that FR is a well-established and common target for cancer due to its overexpression in many cancers, Elechalawar and collaborators [52] encapsulated a compound structurally similar to tamoxifen, bis-arylidene oxindole (Bis-AO, Figure 10), in liposomes containing a new FR-targeting ligand (FA8) by conjugating folic acid and cationic lipid (Table 3, C7). The resulting formulation induced caspase-8 activation and subsequent cleavage of pro-survival factor RIP-1 (Table 4, F5), as well as significant tumor growth inhibition in a murine melanoma model (Table 5, F2). Biodistribution assays were also performed at a dose of 5 mg/kg body weight and a higher accumulation of liposomal formulation in tumor tissue was found, compared to other tissues such as lung, liver, kidney, spleen and heart. In addition, histologic analysis showed that there was no tissue damage or any distinct pathological changes in these vital organs [52]. Lee et al. [53] developed edelfosine-loaded (Figure 10) PEGylated liposomes specifically targeted to integrin αvβ3, which is expressed at higher levels in tumor cells (Table 3, C8). In vitro, this liposomal formulation significantly reduced cell viability of human melanoma cells (A375) compared to non-targeted liposomes (Table 4, F6). The proof of concept in A375 melanoma mice model demonstrated the superior therapeutic efficacy of integrin αvβ3-targeted edelfosine liposomal formulation, with significantly reduced tumor growth, as well as improved survival (Table 5, F3). Following these promising results, the authors further considered optimizing this liposomal formulation with imaging agents for multifunctional purposes [53].

Taking into consideration that a combination of two drugs into the same formulation may provide a synergistic effect, celecoxib and plumbagin (Figure 11) were encapsulated in PEGylated liposomes, designated as Cele-Plum-777 (Table 3, C9) [54]. Cele-Plum-777 displayed the highest in vitro cytotoxic effects against different human melanoma cell lines when comparing to those treated with empty liposomes and those containing celecoxib or plumbagin alone. This liposomal formulation allowed for an optimal drug release profile and, consequently, a maximal synergistic effect both in in vitro (Table 4, F7) and in vivo (Table 5, F4) by diminishing the levels of cyclins involved in tumor cell proliferation and survival. In the in vivo xenograft melanoma model, CelePlum-777 synergistically inhibited tumor growth, with negligible systemic toxicity [54].

Another study focused on the antitumoral agent plumbagin alone, which was incorporated into conventional and long circulating liposomes (Table 3, C10) [183]. The in vitro release profile of plumbagin from liposomal formulations showed an initial burst release, followed by a sustained release phase. As expected, the half-life of the compound was prolonged by PEGylated liposomes, in comparison with conventional liposomes and free plumbagin (elimination half-life of 1305.8 ± 278.2, 346.9 ± 33.8 and 35.9 ± 8.0 min, respectively). In the in vivo murine melanoma model, compared to the free compound, a significantly higher inhibition of tumor progression was achieved for mice treated with PEGylated and conventional liposomes (Table 5, F5). Furthermore, there were no significant changes in hematological parameters upon administration of either free or nanoformulated plumbagin at repeated doses of 2 mg/kg for five consecutive days, indicating its safety. Additional histopathological examination of all treated groups showed no signs of toxic pathological changes in liver, heart, kidney and spleen. Survival studies also revealed an improved life span of 12.5% for group receiving plumbagin-loaded PEGylated liposomes, when compared to free plumbagin [183].

Considering that short-chain ceramides have exhibited interesting antitumor activity, a liposomal C6 ceramide (Figure 12) was evaluated in vitro (Table 4, F8). This liposomal formulation displayed a potent cytotoxicity effect against a panoply of human melanoma cell lines. Moreover, liposomal C6 did not present cytotoxicity towards normal melanocytes, suggesting its selectivity to melanoma cells [55]. In the work of Gowda and collaborators [56], PEGylated liposomes encapsulating arachidonyl trifluoromethyl ketone (ATK) (Figure 12) were studied. ATK is a cytosolic phospholipase A_2_ inhibitor (Table 3, C11) proved to inhibit multiple key pathways involved in recurrent resistant melanoma, as indicated in Table 4 (F9) and Table 5 (F6). No significant differences between ATK in its free or liposomal forms were observed, in in vitro studies, implying that ATK activity is preserved after encapsulation in liposomes. In addition, liposomal ATK was 2 times less toxic to normal control cells compared to melanoma cell lines. In a xenograft melanoma model, nanoformulated ATK showed a dose-dependent reduction of melanoma tumor growth, with a maximum therapeutic efficacy at 30 and 40 mg/kg body weight, with no detectable toxic side effects [56].

The phytosphingosine derivative *N*,*N*,*N*-trimethylphytosphingosine-iodide (3N-TPI, Figure 13), a novel and potent inhibitor of angiogenesis and metastasis, was evaluated in B16F10 murine melanoma cells [184]. This compound was further incorporated in a liposomal formulation, which is described in Table 3 (C12). Interestingly, although in vitro cytotoxicity of liposomal form was much lower than free compound (Table 4, F10), the nanoformulation was able to remarkably suppress in vivo lung tumor metastasis, without major side effects (Table 5, F7) [184]. Another study reported the encapsulation of phosphoethanolamine (PHO-S, Figure 13), a phosphoric ester, into cationic liposomes (Table 3, C13). This nanoformulation demonstrated high cytotoxic activity towards B16F10 melanoma cells, compared with free compound, through induction of G_2_/M-phase cell cycle arrest (Table 4, F11) [185], as well as modulation of the expression of pro-apoptotic proteins (Table 4, F12) [186].

#### 4.2.2. Liposomal Formulations of Natural-Based Compounds

Several clinical antitumoral drugs have been developed from natural sources. In this context, Lin et al. [57] developed liposomes functionalized with α-melanocyte-stimulating hormone (α-MSH) and loading camptothecin (Figure 14) (Table 3, C14). This nanoformulation displayed a high cytotoxicity against B16F10 melanoma cells, with almost a 3-fold decrease in cell viability versus free camptothecin (Table 4, F13). Moreover, α-MSH-targeted liposomes exhibited greater cell endocytosis than non-targeted liposomes and free control [57]. Additionally, a water-soluble analogue of camptothecin, CKD-602 (Figure 14), was also nanoformulated (Table 3, C15) and evaluated against melanoma [58]. In this case, different dosage regimens of liposomal and free CKD-602 were compared in a xenograft melanoma mice model. The nanoformulation was more efficacious than the free drug and the optimal treatment schedule for safe and effective administration was defined as once weekly (Table 5, F8) [58]. In addition, it was found that liposomal compound provided pharmacokinetic advantages in plasma and tumors when compared with free CKD-602 [187]. For instance, the sum total plasma exposure of liposomal CKD602 was 25-fold greater than free compound. Moreover, the concentration-time profile of CKD-602 in tumor extracellular fluid was detectable from 10 min to 75.25 h after liposomes administration, which was significantly greater than the one observed following the administration of CKD-602 in the free form [187].

The compound *n*-butylidenephthalide (BP, Figure 15), isolated from *Angelica sinensis*, was studied due to the fact that several evidence point to the anticancer activity of this compound [188]. In order to improve its biochemical properties, BP was encapsulated in a novel polycationic liposome [liposome-polyethylenimine-polyethylene glycol (LPPC)] (Table 3, C16) and the results demonstrated that this liposomal formulation had higher antitumoral potential than BP alone. Beyond the interesting results exhibited in Table 4 (F14), the combination of liposomal BP and clinical drug 5-Fluorouracil displayed a synergistic cytotoxic effect towards melanoma cells. Moreover, LPPC encapsulation improved the uptake of BP by melanoma cells, by promoting cell endocytosis and maintained BP cytotoxic activity within 24 h [188]. On the other hand, to develop a single nanoparticle-based agent able to target multiple key pathways involved in melanoma development (PI3 kinase, STAT3 and MAP kinase signaling pathways), a potential antitumoral compound, leelamine (Figure 15), was identified from a natural product library [189]. Leelamine was incorporated in a stable PEGylated liposomal delivery system (Table 3, C17), named Nanolipolee-007, presenting a loading efficiency of 64.1% and a release of around 69.1% for 24 h. Liposomal leelamine targeted the signaling pathways referred above (Table 4, F15) and it was 5.69-fold more effective at inducing cell melanoma death, opposed to normal cells. In a xenograft melanoma model (Table 5, F9), Nanolipolee-007 inhibited tumor growth, reducing cell proliferation and tumor vascularization and promoting apoptosis, with negligible toxic effects [189]. Additionally, Hwang et al. [190] isolated the flavonoid anthocyanin from *Hibiscus sabdariffa* Linn. (Table 3, C18). Anthocyanin displays antioxidant and wound-healing properties; however, it is highly susceptible to environmental factors. Thus, the authors encapsulated this molecule into liposomes to enhance its stability. In vitro, this nanoformulation reduced the melanin content of human A375 melanocytes, compared to free anthocyanin. Furthermore, the inhibition of tyrosinase activity by anthocyanin was concentration-dependent and the liposomal formulation significantly improved the inhibitory effect of free anthocyanin (Table 4, F16). Overall, this approach may be suitable for potential applications in melanoma treatment.

Juglone (Figure 15) is a naphthoquinone pigment commonly found in the Juglandaceae family that presents an excellent antitumor potential [59]. Aithal and colleagues [59] have encapsulated juglone in PEGylated liposomes, described in Table 3 (C19). In vitro studies showed an initial burst release, followed by sustained release (around 61% after 24 h). Liposomal juglone exhibited improved pharmacokinetics profile with a 12-fold increase in plasma half-life, comparing with free juglone (Table 4, F17). Further, biodistribution studies performed in the in vivo B16F1 murine melanoma model showed rapid renal elimination of free juglone, proven by its significant accumulation in kidneys. On the contrary, the use of liposomes remarkably reduced juglone accumulation in kidneys, resulting in significantly improved antitumor efficacy (Table 5, F10). Moreover, the histological analysis also revealed lower levels of nephrotoxicity for liposomal juglone compared with those obtained for free juglone [59], rendering this a promising antitumor strategy. In the work of Roseanu and collaborators [191], iron-free lactoferrin was loaded into positively charged liposomes composed by phosphatidyl choline (PC), DOPE, Chol and stearylamine (SA). This liposomal formulation was found to enhance the capacity of lactoferrin to inhibit B16F10 cells proliferation by affecting cell cycle progression (Table 4, F18). Moreover, the internalization of liposomal lactoferrin was increased by 15 to 44% compared to free lactoferrin. The researchers concluded that liposomes are an advantageous tool to improve the antitumor activity of lactoferrin, having a potential therapeutic application in cancer management [191].

Huang and coworkers [60] encapsulated cytochalasin D (CytD, Figure 16) in PEGylated liposomes (Table 3, C20). Biodistribution studies in a B16 murine melanoma model revealed that this liposomal formulation accumulated, to a greater extent, in tumor tissues than free cytD, leading to significant tumor growth inhibition and improved survival rates. Overall, the results (Table 4, F19 and Table 5, F11) indicated that nanoformulated cytD displayed antitumor effects similar to those of the clinical drug cisplatin [60].

The extracts of *natsumikan* from peels of *Citrus natsudaidai* were also incorporated in liposomes and evaluated against murine melanoma cells, with satisfactory results [192]. In general, all liposomal extracts showed improved inhibitory effects in relation to those of free extracts. The nanoformulated extract that displayed the most promising activity was that extracted with petroleum ether. For this reason, the liposome properties of Table 3 (C21) and the in vitro activity in Table 4 (F20) are related to this extract [192].

### 4.3. Other Nanotechnological Systems

Although to a lesser extent, other nanoparticulate systems have been explored as delivery tools for new bioactive molecules against melanoma. An example is given by Nahak and collaborators [193] that loaded ursolic acid in nanostructured lipid carriers (NLCs), prepared by the hot homogenization−ultrasonication method. These NLCs were composed by tribehenin/trierucin:hydrogenated soy phosphatidyl choline:oleic acid/behenic acid, presenting a mean size between 147–235 nm and entrapment efficiencies ranging from 78.4 to 99.9%. All ursolic acid-loaded formulations exhibited superior anticancer activity compared to the free ursolic acid against melanoma cell line B16 (IC_50_ values ranging from 0.041 to 10 μM versus 7.7 μM, respectively) [193].

Orienti et al. [194] developed an oral micellar fenretinide formulation aiming to improve the bioavailability and, consequently, the antitumor efficacy of this synthetic retinoid. This formulation displayed notable antitumor activity against melanoma both in vitro in patient-derived cancer stem cells. Additionally, in lung, colon and melanoma xenografts, a prominent reduction of tumor growth rate was observed at 100 mg/kg, without systemic toxicity. Moreover, pharmacokinetic studies showed that, after oral administration, therapeutic concentrations of the compound were found within tumors [194].

In the work of Athawale and colleagues [195], solid lipid nanoparticles (SLNs) were loaded with etoposide, a hydrophobic semi-synthetic podophyllotoxin derived from *Podophyllum* peltatum roots. In vitro, this nanosystem demonstrated antiproliferative activity against B16F10 murine melanoma cells. In a metastatic melanoma B16F10 model, treatment with etoposide-loaded SLNs resulted in a significant reduction in lung melanoma metastasis, as well as an increased animal survival rate and reduced systemic toxicity, compared to free etoposide [195]. More recently, Valdes and coworkers [196] formulated in SLNs the lipophilic synthetic compound 4-(*N*)-docosahexaenoyl 2′,2′-difluorodeoxycytidine (DHA-dFdC). The use of SLNs improved the solubility of the compounds, chemical stability and cytotoxic activity towards melanoma cells. In a B16F10 melanoma model, the developed nanoformulation effectively reduced tumor growth, compared to free DHA-dFdC, unloaded SLNs, control and vehicle-treated experimental groups [196].

In the case of Bariwal and collaborators [197], the researchers synthesized a new tubulin destabilizing agent—2-(4-hydroxy-1*H*-indol-3-yl)-1*H*-imidazol-4-yl)(3,4,5-trimethoxyphenyl) methanone (QW-296)—which was subsequently formulated in polymeric nanoparticles. In vitro, this nanoparticulate system inhibited melanoma cell proliferation and invasion of B16F10 murine and A375 human melanoma cell lines. In vivo, the systemic administration of QW-296 nanoformulation reduced melanoma tumor growth and significantly inhibited lung melanoma metastasis, compared to the control group [197].

Another research group [198] encapsulated the tryptanthrin derivative CY-1-4, a potential inhibitor of indoleamine 2,3-dioxygenase (IDO), in polycaprolactone-based nanoparticles. The results demonstrated a concentration-dependent inhibition of IDO activity by CY-1-4. Also, both free and encapsulated CY-1-4 displayed low IC_50_ values against HeLa and B16F10 cells and higher IC_50_ towards normal human cells (LX-2; human hepatic stellate cell line), indicating tumor selectivity. In B16F10 tumor-bearing mice, CY-1-4 nanoparticles significantly inhibited tumor growth, with the maximum therapeutic effect achieved at the medium dose of 300 mg/kg [198].

Overall, nanotechnology continuously provides researchers and clinicians with countless opportunities for versatile, advantageous and innovative advancements in the area of melanoma systemic therapy, ultimately benefiting patients.

## 5. Expert Opinion

Conventional pharmacological treatment of melanoma is often accompanied by severe adverse effects, which have a significant negative impact on the quality of life of the patients. Therefore, in the last years, several nanoformulations have been explored for their potential in melanoma treatment with interesting results, as described in this review. Since most of melanoma metastases are not in the skin but rather in vital organs, such as lungs and brain, we focused on systemic therapy instead of transdermal delivery of bioactive molecules, which do not reach metastatic sites.

Taking advantage of the benefits of lipid-based systems, all compounds were nanoformulated in order to overcome problems of solubility, instability and toxicity, which constitute critical obstacles for their therapeutic efficacy. In this context, one important issue is the use of dimethyl sulfoxide as a solubilizing agent for intravenous administration, which can cause high degree of toxicity, rendering this solvent a nonviable option for human use.

Since intravenously administered liposomes can be rapidly cleared from the systemic circulation by the cells of mononuclear phagocytic system, surface modification of liposomes with PEG can increase the circulation half-life of the liposomes. Indeed, PEGylated liposomes are among the most studied nanocarrier systems, as proven in Table 3, where the majority of authors used DSPE-PEG to obtain the most suitable liposomes. In addition, another component often present in the lipid composition is Chol (see Table 3), because it can increase liposomes stability by modulating the fluidity of the lipid bilayer. Besides lipid composition, other parameters also significantly influence the biological activity of the molecules encapsulated/incorporated in liposomes. For instance, liposomes of a smaller size (around 100 nm in diameter) have improved the chances of tumor site accumulation through the EPR effect. Contrarily, liposomes with a bigger size are prone to rapid clearance rates by the mononuclear phagocytic system.

Another important feature is liposome charge. Since tumorigenic cells (either primary cultures or metastases), such as melanoma, express higher levels of negatively charged phospholipids than non-cancerous cells, cationic liposomes can represent an effective strategy for melanoma therapy by electrostatic interactions. This approach has been explored by several authors [182,185,188] to produce promising formulations with anticancer activity. Additionally, the modification of liposome surface with specific ligands is an attractive strategy to enhance therapeutic effects by increasing the recognition and selectivity for tumor cells, as proved by References [52] and [53].

In some cases, liposomal formulations demonstrated lower in vitro cytotoxicity than corresponding free compounds. One plausible explanation is the fact that, once compounds are incorporated/encapsulated in liposomes, they are not immediately available to exert cytotoxic effects. Despite the valuable and substantial information provided by the in vitro models, these assays cannot totally mimic the specific pharmacodynamic actions that occur in the complex in vivo systems. Therefore, ideally, these data should be integrated and interpreted with those obtained from in vivo tests. On the other hand, as showed in Table 5, most studies using xenograft murine melanoma models demonstrated interesting results, such as the reduction of tumor growth or the increase of survival rates when compounds were associated to lipid-based systems. Although syngeneic murine models provide important information about the biological effects of new chemical or natural agents, the employment of a murine metastatic melanoma model is strongly recommended, due to the fact that this model most closely resembles human pathology, enabling the development of more effective therapeutic approaches for this highly aggressive and fatal cancer.

## 6. Conclusions

The increasing incidence and mortality associated with malignant melanoma, as well as the high toxicity and low efficacy of current therapies, prompt the research of novel and improved therapeutic options. To accomplish this demanding goal, the identification of melanoma potential therapeutic targets and the design and development of molecules with antitumor activity are two important tasks. Among the numerous studies, a promising approach has been the synthesis of prodrugs, which have been described as potent and specific inhibitors of melanoma cell growth through tyrosinase. In addition, metal-based compounds have been shown to specifically inhibit AQP3, a transmembrane protein overexpressed in melanoma cells.

Furthermore, the association of these molecules with nanosystems, namely lipid-based ones, is a logical and advantageous strategy for melanoma management. These versatile delivery tools can be tailored according to the objective by means of surface modifications with specific ligand moieties and adaptation of physicochemical properties. The rationale for the use of lipid-based nanosystems, namely liposomes, for the transport and delivery of therapeutic molecules to melanoma sites aims to achieve the minimization of unwanted side effects, a preferential accumulation at diseased tissues and, also, the opportunity to explore tumor biological features for local drug release. With the continuous progress in the nanomedicine field, further advancements are certainly to be expected, providing a paradigm shift in melanoma treatment.

## Figures and Tables

**Figure 1 nanomaterials-09-01455-f001:**
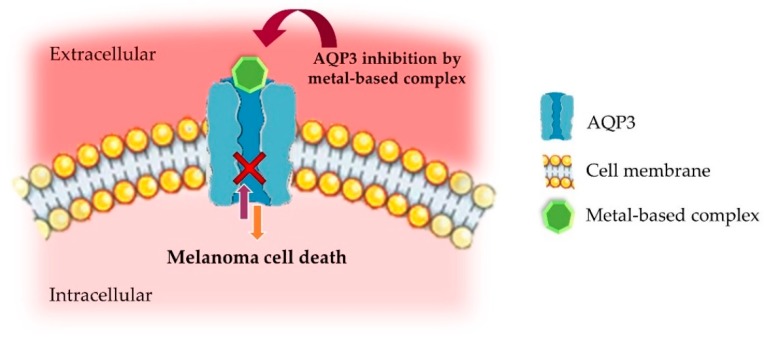
Schematic representation of AQP3 inhibition by metal-based complexes in melanoma cells. Adapted from [28], with permission from Elsevier, 2017.

**Figure 2 nanomaterials-09-01455-f002:**
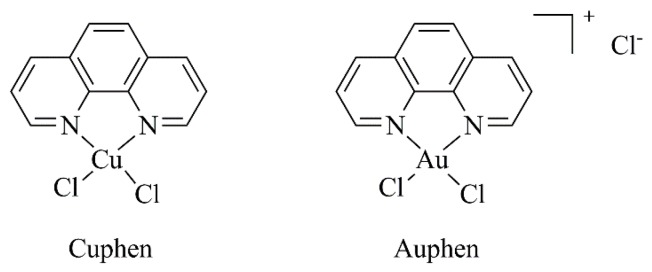
Chemical structures of Cuphen and Auphen.

**Figure 3 nanomaterials-09-01455-f003:**
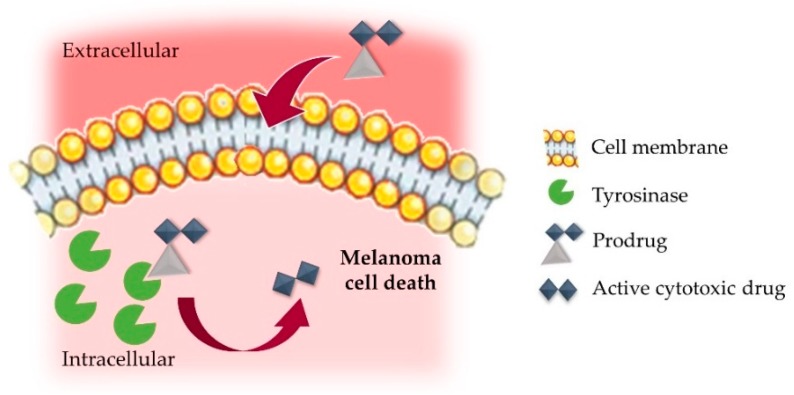
Schematic representation of tyrosinase-mediated prodrug release in a melanoma cell. Adapted from [28], with permission from Elsevier, 2017.

**Figure 4 nanomaterials-09-01455-f004:**
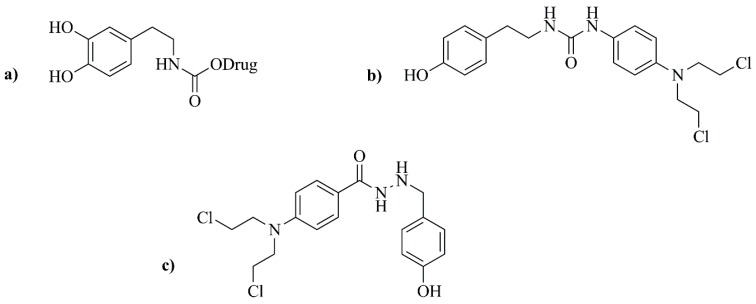
Examples of first-generation prodrugs. (**a**) Prodrug with a dopamine moiety and a carbamate linkage; (**b**) Prodrug with urea-linked aniline mustard; (**c**) Concept of a prodrug displaying a phenolic activator, a hydrazine linker and a nitrogen mustard effector.

**Figure 5 nanomaterials-09-01455-f005:**
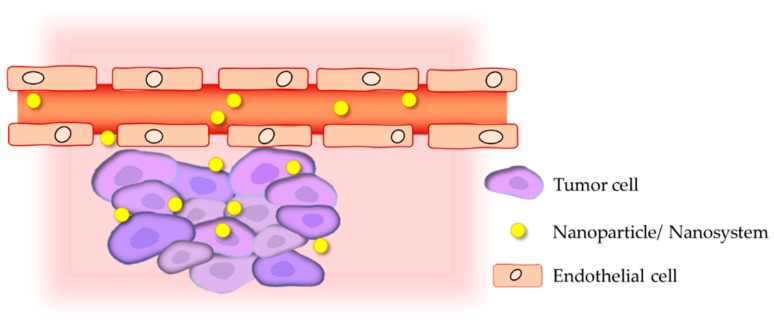
Schematic representation of the Enhanced Permeation and Retention (EPR) effect [158].

**Figure 6 nanomaterials-09-01455-f006:**
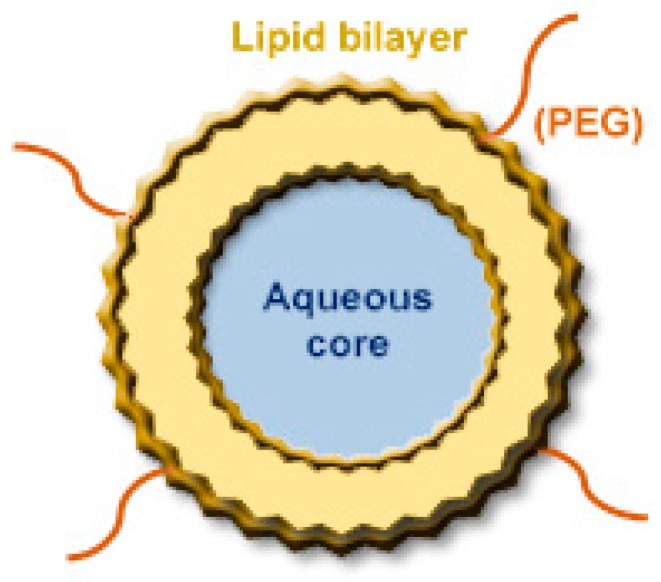
Scheme of a typical PEGylated liposome [158].

**Figure 7 nanomaterials-09-01455-f007:**
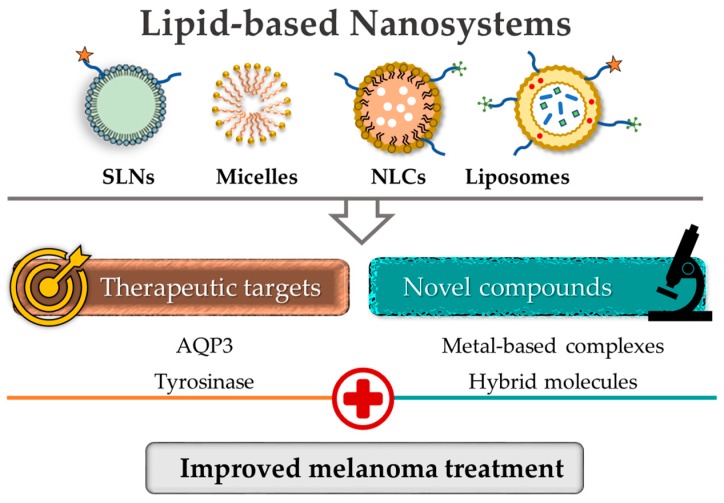
Lipid-based nanosystems as a strategy for melanoma management [158].

**Figure 8 nanomaterials-09-01455-f008:**
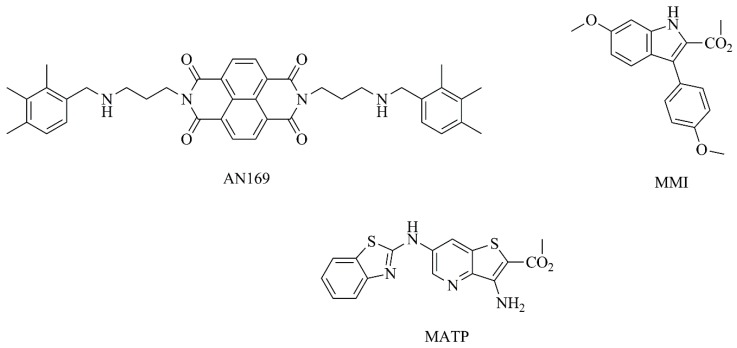
Chemical structures of AN169, MMI and MATP.

**Figure 9 nanomaterials-09-01455-f009:**
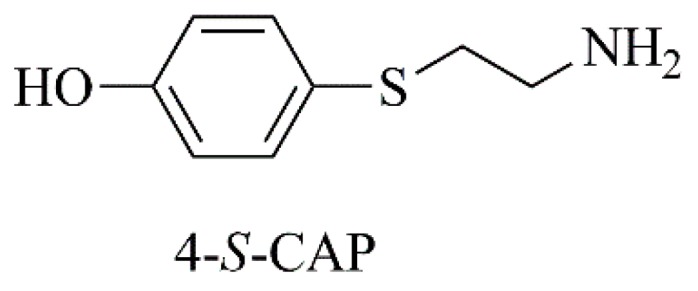
Chemical structure of 4-*S*-CAP.

**Figure 10 nanomaterials-09-01455-f010:**
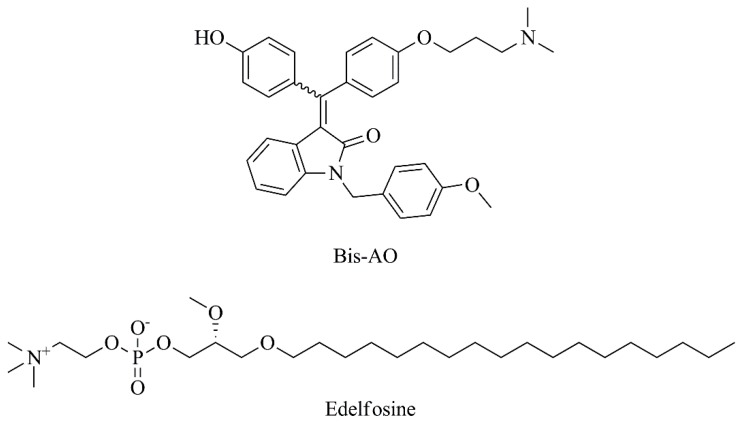
Chemical structures of bis-arylidene oxindole (Bis-AO) and edelfosine.

**Figure 11 nanomaterials-09-01455-f011:**
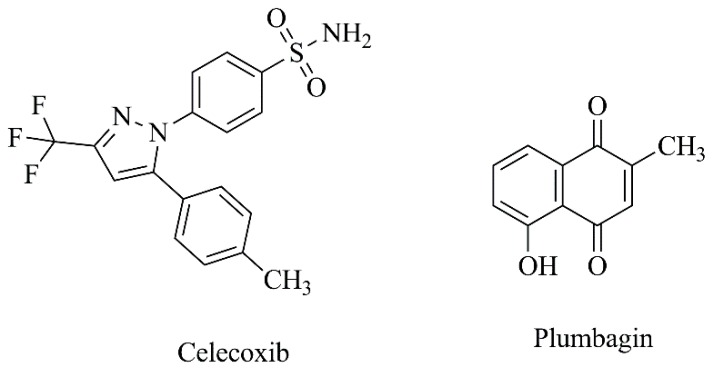
Chemical structures of celecoxib and plumbagin.

**Figure 12 nanomaterials-09-01455-f012:**
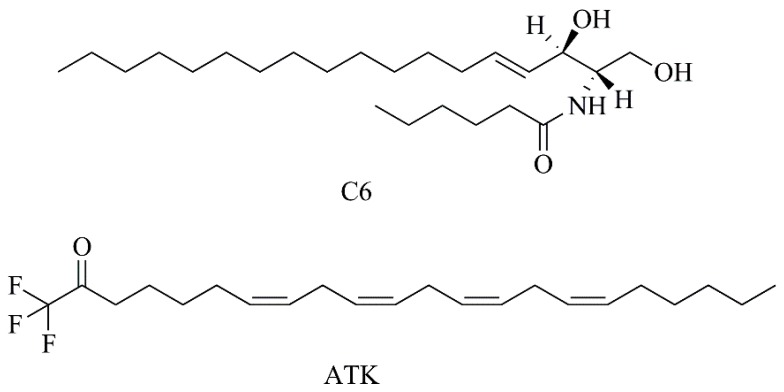
Chemical structures of Ceramide 6 (C6) and ATK.

**Figure 13 nanomaterials-09-01455-f013:**
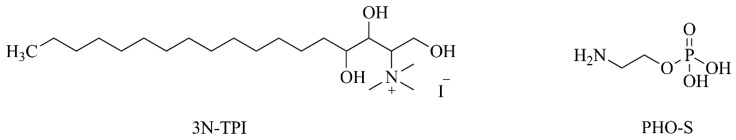
Chemical structures of *N*,*N*,*N*-trimethylphytosphingosine-iodide (3N-TPI) and phosphoethanolamine (PHO-S).

**Figure 14 nanomaterials-09-01455-f014:**
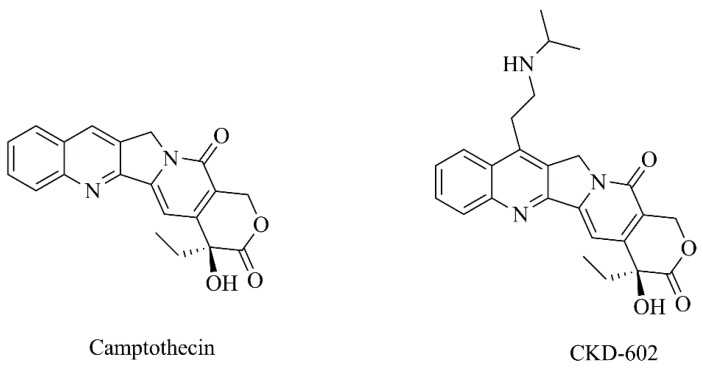
Chemical structures of camptothecin and its derivative, CKD-602.

**Figure 15 nanomaterials-09-01455-f015:**
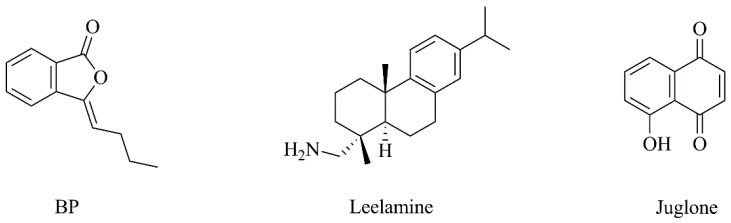
Chemical structures of *n*-butylidenephthalide (BP), leelamine and juglone.

**Figure 16 nanomaterials-09-01455-f016:**
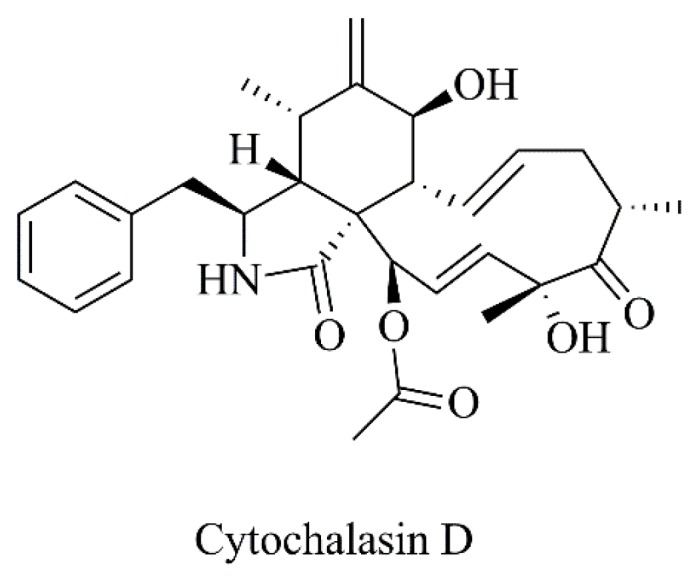
Chemical structure of cytochalasin D.

**Table 1 nanomaterials-09-01455-t001:** Current systemic therapies available for melanoma.

Chemotherapy	Targeted Therapy	Immunotherapy	Combinational
Dacarbazine Temozolamide Carboplatin/cisplatin Vincristine/vinblastine Carmustine/fotemustine	**BRAF inhibitors** (Vemurafenib) (Dabrafenib) (Encorafenib) **MEK inhibitors** (Trametinib) (Cobimetinib) (Binimetinib)	**CTLA-4 mAb** (Ipilimumab) **PD-1 mAb** (Nivolumab) (Pembrolizumab) **Recombinant IFNα2b** **Recombinant IL-2** **T-VEC**	Dabrafenib + trametinib Vemurafenib + cobimetinib Encorafenib + binimetinib Nivolumab + ipilimumab

mAb: monoclonal antibody; CTLA-4: T-lymphocyte-associated protein-4; PD-1: programmed cell death protein-1; IFN-α: interferon alpha; IL-2: interleukin-2; T-VEC: Talimogene laherparepvec; BRAF: serine/threonine protein kinase B-raf; MEK: mitogen activated protein kinase.

**Table 2 nanomaterials-09-01455-t002:** Liposomal formulations approved for clinical use.

Product (Approval Year)	Drug	Lipid Composition	Route	Indication
Abelcet^®^ (1995)	Amphotericin B	DMPC:DMPG	i.v.	Systemic severe fungal infections
Ambisome^®^ (1997)	Amphotericin B	HSPC:DSPG:Chol	i.v.	Presumed fungal infection, Cryptococcal meningitis in HIV patients, visceral leishmaniasis
Amphotec^®^ (1996)	Amphotericin B	Cholesteryl sulfate	i.v.	Invasive aspergillosis
DaunoXome^®^ (1996)	Daunorubicin	DSPC:Chol	i.v.	AIDS-related Kaposi’s sarcoma
Depocyt^®^ (1999)	Cytarabine	DOPC:DPPG:Chol:Triolein	intrathecal	Lymphomatous meningitis
DepoDur^TM^ (2004)	Morphine sulfate	DOPC:Chol:DPPG:tricaprylin, triolein	epidural	Pain management
Doxil^®^/Caelyx® (1995/1996)	Doxorubicin	HSPC:Chol:DSPE-PEG-2000	i.v.	Ovarian and breast cancer, multiple myeloma, AIDS-related Kaposi’s sarcoma
Epaxal® (1993)	Inactivated hepatitis A virus (strain RG-SB)	DOPC:DOPE	i.m.	Hepatitis A
Exparel^®^ (2011)	Bupivacaine	DEPC:DPPG:Chol:tricaprylin	i.v.	Pain management
Marqibo^®^ (2012)	Vincristine	Sphingomyelin:Chol	i.v.	Acute lymphoblastic leukaemia
Mepact^®^ (2004)	Mifamurtide	DOPS:POPC	i.v.	High-grade, resectable, non-metastatic osteosarcoma
Myocet^®^ (2000)	Doxorubicin	PC:Chol	i.v.	Metastatic breast cancer
Onivyde™ (2015)	Irinotecan	DSPC:Chol, DSPE-PEG-2000	i.v.	Metastatic adenocarcinoma of the pancreas
Onpattro^TM^ (2018)	Transthyretin-directed small interfering RNA	DSPC:Chol:DLin-MC3-DMA:DMPG-PEG-2000	i.v.	Polyneuropathy of hereditary transthyretin-mediated amyloidosis
Visudyne^®^ (2000)	Verteporfin	PG:DMPC	i.v.	Subfoveal choroidal neovascularization
Vixeos^TM^ (2017)	Daunorubicin and cytarabine	DSPC:DSPG:Chol	i.v.	Acute myeloid leukemia

PC: phosphatidyl choline; PG: phosphatidyl glycerol; DMPC: dimyristoyl phosphatidyl choline; DMPG: dimyristoyl phosphatidyl glycerol; DSPC: distearoyl phosphatidyl choline; DEPC: dierucoyl phosphatidyl choline; DOPC: dioleoyl phosphatidyl choline; DSPG: distearoyl phosphatidyl glycerol; HSPC: hydrogenated soy phosphatidyl choline; DOPE: dioleoyl phosphatidyl ethanolamine; DOPS: dioleoyl phosphatidyl serine; DPPG: dipalmitoyl phosphatidyl glycerol; POPC: palmitoyl oleoyl phosphatidyl choline; Chol: cholesterol; DSPE-PEG-2000: distearoyl phosphatidyl ethanolamine covalently linked to polyethylene glycol-2000; DMPG-PEG-2000: dimyristoyl phosphatidyl glycerol covalently linked to polyethylene glycol-2000; DLin-MC3-DMA: (6Z,9Z,28Z,31Z)-heptatriaconta-6,9,28,31-tetraen-19-yl-4-(dimethylamino) butanoate; i.v.: intravenous; i.m.: intramuscular; RNA: ribonucleic acid; AIDS: acquired immunodeficiency syndrome.

**Table 3 nanomaterials-09-01455-t003:** Physicochemical properties of the most suitable liposomal formulations for different compounds with antitumor activity.

Compound (No/Name)	The Best Formulation	Size (nm)	PdI	Zeta Potential (mV)	EE/IE (%)	Reference
C1/Cuphen	PC:Chol:DSPE-PEG-2000	160	<0.15	−4 ± 1	47 ± 5	[97]
C2/Cuphen	DMPC:CHEMS:DSPE-PEG-2000	130	<0.10	−3 ± 1	86 ± 7	[101]
C3/AN169	HSPC:Chol:DSPE-PEG-2000	147 ± 7	0.08 ± 0.02	-	87 ± 3	[178]
C4/MMI	PC:Chol:DPPG	104 ± 1	0.12 ± 0.01	−52 ± 6	98	[179]
	PC:DPPG:DSPE-PEG-2000	104 ± 3	0.27 ± 0.01	−43 ± 3	99	
C5/MATP	PC:Chol	84 ± 2	0.23 ± 0.004	-	-	[180]
	PC:Chol:DMPG	64 ± 0.4	0.28 ± 0.003	-	-	
	DPPC:DMPG	91 ± 1	0.23 ± 0.01	-	-	
C6/4-*S*-CAP	Magnetite cationic TMAG:DLPC:DOPE	~400	-	-	-	[182]
C7/Bis-AO	DOPC:Chol:Cationic folate	167 ± 8	0.32	12 ± 3	-	[52]
C8/Edelfosine	PC:PG:Chol:DSPE-PEG-2000 or tetrac-DSPE-PEG-2000	193 ± 4	0.17 ± 0.03	-	-	[53,199]
C9/Celecoxib + Plumbagin	PC:DPPE-PEG-2000	71	-	−1 ± 0.4	89 (celecoxib), 68 (plumbagin)	[54]
C10/Plumbagin	PC:Chol	115 ± 7	0.27	−63	-	[183]
	PC:Chol:DSPE-PEG-2000	118 ± 1	0.23	−56	67 ± 2	
C11/ATK	PC:DPPE-PEG-2000	68 ± 6	-	−0.4 ± 0.04	62	[56]
C12/3N-TPI	DPPC:Chol	152 ± 7	0.12 ± 0.05	−0.3	-	[184]
C13/PHO-S	DODAC	152	-	56 ± 8	51	[185]
C14/Camptothecin	α-MSH-PC:Chol:SA	253 ± 6	0.24 ± 0.02	60 ± 1	95 ± 0.3	[57]
C15/CKD-602	DSPE-PEG:DSPC	100	-	-	96	[58]
C16/BP	DOPC:DLPC:PEG:PEI	200–280	-	~38	-	[188,200]
C17/Leelamine	PC:DPPE-PEG-2000	67	-	0.1	-	[189]
C18/Anthocyanin	Lecithin:Chol:other lipids	156 ± 1	-	-	55 ± 3	[190]
C19/Juglone	PC:Chol:DSPE-PEG-2000	117 ± 2	0.23	−32	67 ± 3	[59]
C20/CytD	Lecithin:Chol:PEG-4000	150 ± 30	-	-	-	[60]
C21/Extracts of *natsumikan*	DMPC:Tween 20	90	-	-	-	[192]

Abbreviations: α-MSH: α-melanocyte-stimulating hormone; CHEMS: cholesteryl hemisuccinate; Chol: Cholesterol; DLPC: dilauroyl phosphatidyl choline; DMPC: dimyristoyl phosphatidyl choline; DMPG: dimyristoyl phosphatidyl glycerol; DODAC: dioctadecyl dimethylammonium chloride; DOPC: dioleoyl phosphatidyl choline; DOPE: dioleoyl phosphatidyl ethanolamine; DPPC: Dipalmitoyl phosphatidyl choline; DPPE-PEG: dipalmitoyl phosphatidyl ethanolamine covalently linked to polyethylene glycol; DPPG: dipalmitoyl phosphatidyl glycerol; DSPC: distearoyl phosphatidyl choline; DSPE-PEG: distearoyl phosphatidyl ethanolamine covalently linked to polyethylene glycol; PEI: poly ethylenimine; PG: phosphatidyl glycerol; SA: stearylamine; TMAG: trimethyl ammonioacetyl glutamate; EE: encapsulation efficiency; HSPC: hydrogenated soy phosphatidyl choline; IE: incorporation efficiency; PC: phosphatidyl choline; PdI: polydispersity index.

**Table 4 nanomaterials-09-01455-t004:** In vitro evaluation studies of the different compound-loaded liposomes.

Formulation/Compound (No/Name)	Assay (Cell Lines)	Main Results	Reference
F1/Cuphen	MTS assay (MNT-1 and B16F10)	IC_50_ = 4.4 ± 0.2 µM (MNT-1) and 5.1 ± 0.1 µM (B16F10) versus 3.1 ± 0.2 µM (MNT-1) and 3.3 ± 0.3 µM (B16F10) for free Cuphen	[97]
	Flow cytometry (MNT-1)	Loss of cell viability = 80%	
	Hemolytic activity assay	Hemolysis < 4%	
F2/Cuphen	MTS assay (B16F10)	IC_50_ = 2.6 ± 0.9 µM versus 3.4 ± 0.6 µM for free Cuphen	[101]
F3/AN169	MTT assay (Mel 3.0)	IC_50_ = 0.8 ± 0.01 µM versus 0.75 ± 0.04 µM for free compound	[178]
F4/4-*S*-CAP	Trypan blue dye-exclusion method (B16)	Relative cell number = 46.6 ± 0.9% (400 μM)	[182]
F5/Bis-AO	MTT (B16F10, A549, SKOV-3 and NIH3T3)	Cell viability around 18%, 85%, 22% and 85%, respectively (10 µM)	[52]
	Flow cytometry (B16F10)	↑ Necrotic cells accumulation	
	Western Blot (B16F10)	Up-regulation of RIP1 cleaved fragments Up-regulation of caspase-8	
F6/Edelfosine	MTT assay (A375)	↑ Tumor cells death (to 48.0 ± 4.1%)	[53]
F7/Celecoxib + Plumbagin	MTS assay (UACC 903 and 1205 Lu)	Cell viability ~25%	[54]
	Western Blot (UACC 903 and 1205 Lu)	↑ COX-2 levels ↓ Protein levels of pSTAT3 (Y705) ↓ Cyclins B1, D1 and H Induction of caspase-3/7 activity	
F8/C6	MTT assay (WM-115, SK-Mel2, WM-266.4 and A-375)	Cell survival ~40% (WM-115), ~35% (SK-Mel2), ~10% (WM-266.4) and ~55% (A-375) (10 µM)	[55]
	Colorimetric assay (WM-115)	↑ Activity of caspse-3 and caspase-9	
	Flow cytometry and ELISA assay (WM-115)	↑ Annexin V percentage and ssDNA ELISA OD	
	Western Blot (WM-115 and A-375)	↑ Protein phosphatase activity (PP1) Inhibition of Akt-mTOR signaling	
F9/ATK	Hemolytic activity assay	0.55% hemolysis versus arachidonyl trifluoromethyl ketone dissolved in ethanol (2.9%)	[56]
	MTS assay (UACC 903 and 1205 Lu)	IC_50_ = 20 μmol/L	
	Western Blot (UACC 903 and 1205 Lu)	↑ Caspase-3/7 activity ↓ Levels of cyclin D1 and cPLA_2_ activity ↑ p21 and p27, cleaved PARP and LC3B and COX-2 protein expression ↓ Levels of pAKT and pSTAT3	
F10/3N-TPI	MTT assay (A375P and B16F10)	IC_50_ = 372.5 ± 42.5 µM (A375P cells) and >400 µM (B16F10 cells) versus 13.0 ± 0.5 µM (A375P cells) and 33.3 ± 1.0 (B16F10 cells) for free compound	[184]
	Hemolytic activity assay	No hemolysis at <2 mM	
	Cell migration assay (B16F10)	↓ Wound healing of about 40–50% (100, 200 and 400 µM)	
	Western Blot (B16F10)	Inhibition of VEGF and MMP-2 activity	
F11/PHO-S	MTT assay (B16F10)	IC_50_ = 0.8 mM versus 4.4 mM for free form	[185]
	Flow cytometry (B16F10)	↑ Population of cells in the G_2_/M phase (20.5 ± 1.2% versus 14.4 ± 1.3% for free compound)	
F12/PHO-S	Flow cytometry (B16F10)	↑ TRAIL-DR4 receptor expression 8.4 ± 0.4% Modulation of the expression of caspases 3 (11.7 ± 0.3%) and 8 (29.8 ± 5.5%) at 2 mM ↑ Free cytochrome c in the cytoplasm (4.4 ± 0.6%) at 2 mM	[186]
F13/Camptothecin	Bioluminescence assay (B16F10)	Cell viability = 18% versus 32% of non-targeted liposome and 48% of free camptothecin (50 µM)	[57]
F14/BP	MTT assay (B16F10, K-balb)	IC_50_ = 12.2 and 15.3 µg/mL, respectively	[188]
	Flow cytometry (B16F10)	↑ Cell cycle arrest at G_0_/G_1_ phase ↓ Protein expression of RB, p-RB, CDK4 and cyclin D1 ↑ Protein expression of P53, p-P53 and P21	
	TUNEL assay (B16F10)	Chromatin condensation DNA fragmentation Apoptotic bodies	
	Immunocytochemistry	Activation of Fas, FasL, Cleaved-Cas-8 Activation of Bax, AIF, Cleaved-Cas-9	
	Western Blot (B16F10)	Activation of caspase-3, -8 and -9	
F15/Leelamine	Hemolytic activity assay	3.3% hemolysis versus leelamine dissolved in DMSO (15.8%)	[189]
	MTS assay (UACC 903 and 1205 Lu)	IC_50_ = 2.3 µM	
	ELISA assay (UACC 903 and 1205 Lu)	↓ Cellular proliferation ↑ Cellular apoptosis	
	Flow cytometry (UACC 903 and 1205 Lu)	↑ Sub-G_0_/G_1_ and G_0_/G_1_ cell populations	
	Western Blot (UACC 903 and 1205 Lu)	↓ Activity of PI3K/Akt, STAT3 and MAPK Inhibition of Akt phosphorylation ↓ Expression of cyclin D1 ↑ Cleaved caspase-3 and PARP protein levels Inhibition of phosphorylation of STAT3	
F16/Anthocyanin	DPPH assay	Radical-scavenging activity = 64 and 76% (20 and 50 mg/mL, respectively)	[190]
	MTT assay (A375)	Cell viability = 80% (200 mg/mL)	
	Melanin content assay (A375)	↓ Melanin production (inhibitory effect of 60% versus 30% of free anthocyanin at 50 mg/mL)	
	Cellular tyrosinase assay (A375)	Inhibition of tyrosinase activity (58% versus 30% of free anthocyanin at 50 mg/mL)	
	Western Blot (A375)	Inhibition of tyrosinase and MITF expression	
F17/Juglone	MTT assay (B16F10)	IC_50_ = 4.1 µM versus 7.8 µM for free compound	[59]
F18/Lactoferrin	MTS assay (B16F10)	↓ Cell viability 10–15% regarding free compound	[191]
	Flow cytometry (B16F10)	↑ Cell cycle arrest at G_0_-G_1_ phase	
F19/CytD	MTT assay (B16)	Relative inhibition = 73.3 ± 8.9% (7.5 μg/mL)	[60]
	TUNEL assay (B16)	Induction of cell apoptosis	
F20/*natsumikan* extracts	WST-1 (B16)	↑ Inhibitory effect comparing to free extract	[192]
	Fluorescence microscopic assay (B16)	Induction of apoptosis	

**Table 5 nanomaterials-09-01455-t005:** In vivo proof of concept studies of nanoformulated compounds in melanoma murine models.

Formulation/Compound (No/Name)	Animals (*n*)	Animal Model	Treatment (Dose)	Effects	Reference
F1/Cuphen	C57Bl/6 mice (5)	Syngeneic melanoma model (B16F10)	i.v., three-times a week, for 2 weeks (2.5 mg/kg)	Delay of tumor progression RTV = 8 versus 13 for free Cuphen and 24 for control	[101]
F2/Bis-AO	C57BL/6J mice	Syngeneic melanoma model (B16F10)	i.v., five injections every alternate day	↓ Tumor volume ↑ Survivability	[52]
F3/Edelfosine	Athymic nude mice (4)	Syngeneic melanoma model (A375)	i.v. on days 9, 11, 13, 15 and 17 (20 mg/kg)	↓ Melanoma tumor growth (169.5 ± 64.6 mm^3^ on day 30) ↑ Survival time (54 days versus 48 days of edelfosine in pegylated liposome)	[53]
F4/Celecoxib + Plumbagin	Athymic-foxn1^nu^ nude mice	Syngeneic melanoma model (UACC 903 or 1205 Lu)	i.v., alternate day, 3–4 weeks (15 + 1.5mg/kg)	Tumor inhibition up to 72%	[54]
F5/Plumbagin	C57BL/6J mice (8)	Syngeneic melanoma model (B16F10)	i.v., on days 1, 3, 5, 7 and 10 (2 mg/kg)	↓ Tumor volume (VDT = 4.2 ± 0.7 for pegylated liposomes; 3.8 ± 0.6 for conventional liposomes versus 2.4 ± 0.2 for free plumbagin and 1.6 ± 0.4 for vehicle-treated animals)	[183]
F6/ATK	Athymic-foxn1^nu^ nude mice (4)	Syngeneic melanoma model (UACC 903 or 1205 Lu)	i.v., daily, 3–4 weeks (30 and 40 mg/kg)	↓ Melanoma tumor growth [58% (UACC 903) and 55% (1205 Lu)]	[56]
F7/3N-TPI	C57BL/6 mice	Metastatic model (B16F10)	15 min, 5 and 10 days after tumor inoculation (0.4 and 2 mM)	↓ Number of lung nodules, compared to vehicle control and free compound	[184]
F8/CKD-602	NCR.nu/nu homozygous mice (5–12)	Syngeneic melanoma model (A375)	i.v., once weekly, twice weekly or once every 2 weeks, for 3 weeks (0.1 to 3.5 mg/kg)	CTR = ≥ 0.3 mg/kg (once weekly administration) MED = 0.15 mg/kg (once weekly), ≤ 0.3 mg/kg (twice weekly) and 0.1–0.3 mg/kg (every 2 weeks) versus ≤30 mg/kg for free form (once weekly) TI = 10 (once weekly), ~8 (twice weekly) and ~5 (every 2 weeks) versus >1 for free form (once weekly)	[58]
F9/Leelamine	Athymic-foxn1^nu^ nude mice (5)	Syngeneic melanoma model (UACC 903 or 1205 Lu)	i.v., daily, 3–4 weeks (30 mg/kg)	↓ Tumor volume (~55%)	[189]
F10/Juglone	C57BL/6J mice	Syngeneic melanoma model (B16F1)	i.v. on days 1, 3 and 5 (1 mg/kg)	Delay tumor growth kinetic parameters (VDT = 3.6 ± 0.7 versus 2.9 ± 0.7 for free juglone and 1.6 ± 0.5 for vehicle control) ↑ Survival time (32 days versus 28 days for free julone and 19 days for vehicle control)	[59]
F11/CytD	C57BL/6N mice (5)	Syngeneic melanoma model (B16)	i.v., every 3 days for 15 days (50 mg/kg)	Inhibition of tumor growth ↑ Survival time ↓ Average number of vessels per high-power field Inhibition of angiogenesis	[60]

Abbreviations: RTV: relative tumor volume; VDT: volume doubling time; MED: minimum efficacious dose; CTR: Complete tumor regression; TI: Therapeutic index—defined as the ratio of the maximum tolerated dose to the minimum efficacious dose.

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
