# Peer review of "Emergent Nanotechnological Strategies for Systemic Chemotherapy against Melanoma"

_nanomaterials, 2019, doi:10.3390/nano9101455_

Round 1

Reviewer 1 Report

In the paper “Emergent Nanotechnological Strategies for Systemic Chemotherapy against Melanoma”, the authors review a huge quantity of information.

After a careful introduction regarding main clinical strategies used against melanoma, the authors underline the need of new drugs and they suggest to use aquaporin 3 and tyrosinase as theurapetic targets. Finally they extensively and slavishly analyze the possibility to use liposomes to carry drugs against melanoma mentioning a long list of papers emphasizing the potential of liposomes use.

This paper is well structured and can be accepted after the following small amendments:

Minor points:

The Authors could introduce a short paragraph on other nanoparticles, in alternative to liposomes, used to carry drugs against melanoma, reporting best results achieved with these different treatments This paper could become more fluent in reading by removing or moving in an additional section, all chemical structural formulas listed for all chemical compounds mentioned. On row 42, in the introduction paragraph, it has to be better specified that it is specific drugs as Vinca alcaloids and taxans to interphere with microtubules functions. On row 81 the sentence (Error! Reference source not found.) should be amended. On row 93 it should specified that the systemic therapy is applied to metastatic melanoma or locally advanced (stage IV and III) On row 148 could be mentioned and introduced more molecules to be used as target in melanoma’s therapy, also referring to the same ones presented in the following section coupled with liposomes.

Author Response

In the paper “Emergent Nanotechnological Strategies for Systemic Chemotherapy against Melanoma”, the authors review a huge quantity of information.

After a careful introduction regarding main clinical strategies used against melanoma, the authors underline the need of new drugs and they suggest to use aquaporin 3 and tyrosinase as therapeutic targets. Finally, they extensively and slavishly analyze the possibility to use liposomes to carry drugs against melanoma mentioning a long list of papers emphasizing the potential of liposomes use.

This paper is well structured and can be accepted after the following small amendments:

Minor points:

The Authors could introduce a short paragraph on other nanoparticles, in alternative to liposomes, used to carry drugs against melanoma, reporting best results achieved with these different treatments

Answer: The section 4.3 was reformulated:

4.3. Other nanotechnological systems

Although in a less extent, other nanoparticulate systems have been explored as delivery tools for new bioactive molecules against melanoma. An example is given by Nahak and collaborators [192] that loaded ursolic acid in nanostructured lipid carriers (NLCs), prepared by the hot homogenization−ultrasonication method. These NLCs were composed by tribehenin/trierucin:hydrogenated soy phosphatidyl choline:oleic acid/behenic acid, presenting a mean size between 147-235 nm and entrapment efficiencies ranging from 78.4 to 99.9%. All ursolic acid-loaded formulations exhibited superior anticancer activity compared to those of free ursolic acid against melanoma cell line B16 (IC50 of 0.041-0.10 μM versus 7.7 μM, respectively) [192].

Orienti et al. [193] developed an oral micellar fenretinide formulation aiming to improve the bioavailability and, consequently, the antitumor efficacy of this synthetic retinoid. This formulation displayed a notable antitumor activity against melanoma both in vitro in patient-derived cancer stem cells. Additionally, in lung, colon and melanoma xenografts, a prominent reduction of tumor growth rate was observed at 100 mg/kg, without systemic toxicity. Moreover, pharmacokinetic studies showed that, after oral administration, therapeutic concentrations of the compound were found within tumors [193].

In the work of Athawale and colleagues [194], solid lipid nanoparticles (SLNs) were loaded with etoposide, a hydrophobic semi-synthetic podophyllotoxin derived from Podophyllum peltatum roots. In vitro, this nanosystem demonstrated antiproliferative activity against B16F10 murine melanoma cells. In a metastatic melanoma B16F10 model, treatment with etoposide-loaded SLNs resulted in a significant reduction in lung melanoma metastasis, as well as an increased animal survival rate and reduced systemic toxicity, compared to free etoposide [194]. More recently, Valdes and coworkers [195] formulated in SLNs the lipophilic synthetic compound 4-(N)-docosahexaenoyl 2′,2′-difluorodeoxycytidine (DHA-dFdC). The use of SLNs improved the solubility of the compounds, chemical stability and cytotoxic activity towards melanoma cells. In a B16F10 melanoma model, the developed nanoformulation effectively reduced tumor growth, comparing to free DHA-dFdC, unloaded SLNs, control and vehicle-treated experimental groups [195].

In the case of Bariwal and collaborators [196], the researchers synthesized a new tubulin destabilizing agent, 2-(4-hydroxy-1H-indol-3-yl)-1H-imidazol-4-yl)(3,4,5-trimethoxyphenyl) methanone (QW-296), which was subsequently formulated in polymeric nanoparticles. In vitro, this nanoparticulate system inhibited melanoma cell proliferation and invasion of B16F10 murine and A375 human melanoma cell lines. In vivo, the systemic administration of QW-296 nanoformulation reduced melanoma tumor growth and significantly inhibited lung melanoma metastasis, comparing to control group [196].

Another research group [197] encapsulated the tryptanthrin derivative CY-1-4, a potential inhibitor of indoleamine 2,3-dioxygenase (IDO), in polycaprolactone-based nanoparticles. The results demonstrated a concentration-dependent inhibition of IDO activity by CY-1-4. Also, both free and encapsulated CY-1-4 displayed low IC50 values against HeLa and B16F10 cells and higher IC50 towards normal human cells (LX-2; human hepatic stellate cell line), indicating tumor selectivity. In B16F10 tumor-bearing mice, CY-1-4 nanoparticles significantly inhibited tumor growth, with the maximum therapeutic effect achieved at the medium dose of 300 mg/kg [197].

Overall, nanotechnology continuously provides researchers and clinicians with countless opportunities for versatile, advantageous and innovative advancements in the area of melanoma systemic therapy, ultimately benefiting patients.

The newly added references (193-197) in the revised manuscript are highlighted in yellow in the list of references.

This paper could become more fluent in reading by removing or moving in an additional section, all chemical structural formulas listed for all chemical compounds mentioned.

Answer: We thank the suggestion regarding the removal or change in an additional section of all the chemical structural formulas. However, in our opinion, we consider that the images provide a more interesting and clear reading, captivating the readers’ attention. Moreover, maintaining the chemical structures close to the respective information avoids the readers to search the structures in other section of the manuscript.

On row 42, in the introduction paragraph, it has to be better specified that it is specific drugs as Vinca alcaloids and taxans to interfere with microtubules functions.

Answer: Thank you for your comment. The information was included in the revised manuscript and a new reference was added (4) - highlighted in yellow in the list of references.

“The standard chemotherapeutic agents preferentially act on dividing cells by inducing DNA damage and strand breakage, which interferes with DNA repair and microtubule function, specifically vinca alkaloids and taxanes [3,4]”.

On row 81 the sentence (Error! Reference source not found.) should be amended.

Answer: The sentence was corrected.

On row 93 it should be specified that the systemic therapy is applied to metastatic melanoma or locally advanced (stage IV and III).

Answer: The section – “Surgery”rows 89 to 96 was corrected accordingly:

“When the disease is early detected (stages I and II), the surgical removal of melanoma can be successfully achieved, with relatively low morbidity [21]. This medical procedure may prevent the occurrence of metastasis; however, in most cases of advanced melanoma, the cancer cannot be eradicated through this approach. Notwithstanding, clinical trials combining surgical resection with systemic therapies have been conducted in melanoma at stages III and IV [21,22].”

An additional reference was added (22), highlighted in yellow in the list of references.

On row 148 could be mentioned and introduced more molecules to be used as target in melanoma’s therapy, also referring to the same ones presented in the following section coupled with liposomes.

Answer: A paragraph was included in the revised manuscript:

“Several targets have been associated to melanoma pathogenesis, such as tyrosinase [47–49], aquaporin-3 (AQP3) [50,51], folate receptor (FR) [52], integrin αvβ3 [53], cyclooxygenase-2 [54], STAT3 [54], protein phosphatase 1 [55], cytosolic phospholipase A2 [56], melanocotin-1 receptor [57], topoisomerase 1 [58], prolyl isomerase Pin1 [59] and actin microfilaments [60]. The use of nanotechnological tools for the modulation of these therapeutic targets will be addressed in this manuscript. In addition, in the context of putative targets, an emphasis will be given in the following sections to AQP3 and tyrosinase, which are up-regulated in melanoma.”

Reviewer 2 Report

The manuscript “Emergent nanotechnological strategies for systemic chemotherapy against melanoma” by Jacinta Oliveira Pinho et al., is an interesting review regarding the strategies for melanoma treatment. The authors report the findings of studies that concern the nanotechnological therapeutic approaches for antitumor activity against melanoma.

The review is well written, and the references are appropriate and up to date.

There are no major issues within the manuscript.

Minor issues:

- The authors should insert Table 1 in the text, maybe it could be included in paragraph 2.1

- In Fig 1 should be shown the metal-based complex, as indicated

- In Fig 3 the symbol AQP should be eliminated because it does not appear in Figure

- In paragraph 3.2 Therapeutic potential of metal-based compounds, it would be interesting to insert some data concerning Ru-containing complexes as drugs with anticancer activity (see for example Riccardi et al., EurJOC 2017; 7, 1099-1119; Piccolo et al., Sci Rep. 2019 9(1):7006).

Author Response

Comments and Suggestions for Authors

The manuscript “Emergent nanotechnological strategies for systemic chemotherapy against melanoma” by Jacinta Oliveira Pinho et al., is an interesting review regarding the strategies for melanoma treatment. The authors report the findings of studies that concern the nanotechnological therapeutic approaches for antitumor activity against melanoma.

The review is well written, and the references are appropriate and up to date.

There are no major issues within the manuscript.

Minor issues:

The authors should insert Table 1 in the text, maybe it could be included in paragraph 2.1

Answer: Table 1 was inserted in the text – line 82.

In Fig 1 should be shown the metal-based complex, as indicated

Answer: Figure 1 was changed accordingly; the metal-based complex was included.

In Fig 3 the symbol AQP should be eliminated because it does not appear in Figure

Answer: The symbol AQP was eliminated in Figure 3.

In paragraph 3.2 Therapeutic potential of metal-based compounds, it would be interesting to insert some data concerning Ru-containing complexes as drugs with anticancer activity (see for example Riccardi et al., EurJOC 2017; 7, 1099-1119; Piccolo et al., Sci Rep. 2019 9(1):7006).

Answer: The information regarding the therapeutic potential of Ru-containing complexes was included in paragraph 3.2:

“Additionally, ruthenium (Ru) complexes have emerged as promising second-generation metal-based anticancer agents. Moreover, some of them have entered in clinical trials. Particularly, Ru(III)-containing nucleolipids showed a remarkable in vitro anticancer activity [90]. More recently, Ru(III)-complexes incorporated into a DOTAP liposomal formulation demonstrated effective antitumor activity both in vitro and in vivo [91].”

The newly added references (90, 91) are highlighted in yellow in the list of references.

Reviewer 3 Report

Melanoma is one of the most difficult tumors to treat, especially at the advanced stages. This review gives overview of melanoma treatment, discusses potential targets, and in particular, compares nanotechnological strategy for effective delivery of selective drugs. The review is well written and comprehensive with updated references. However, there are several minor points that require improvement.

Points

1) Line 81, "(Error! Reference source not found.)": What is this?

2) Table 1, The heading should be the sequence that appears in the main text: Chemotherapy, Targeted Therapy, Immunotherapy, and Combinational. 

3) Line 167, "is small non-polar solutes, such as glycerol and urea": Is this correct? I believe that glycerol and urea are polar solutes. 

4) Figure 1: This should be improved. For example, Metal-based comple is not found in the figure.

5) Lines 388-388: The sentence is strange to me in grammar. 

6) Figure 9: The structure of 4-S-CAP is not correct (one more methylene group is needed). 

Author Response

Reviewer 3

Comments and Suggestions for Authors

Melanoma is one of the most difficult tumors to treat, especially at the advanced stages. This review gives overview of melanoma treatment, discusses potential targets, and in particular, compares nanotechnological strategy for effective delivery of selective drugs. The review is well written and comprehensive with updated references. However, there are several minor points that require improvement.

Points

1. Line 81, "(Error! Reference source not found.)": What is this?

Answer: The sentence was corrected

2. Table 1, The heading should be the sequence that appears in the main text: Chemotherapy, Targeted Therapy, Immunotherapy, and Combinational. 

Answer: Table 1 was changed according to your suggestion.

3. Line 167, "is small non-polar solutes, such as glycerol and urea": Is this correct? I believe that glycerol and urea are polar solutes. 

Answer: Thank you for your comment. You are absolutely correct, both glycerol and urea are polar solutes. The sentence was corrected:

“Aquaporin channels can be grouped into two main categories: orthodox (AQPs 0, 1, 2, 4, 5, 6 and 8), which are water-specific channels, and aquaglyceroporins (AQPs 3, 7, 9, 10), involved in the bidirectional transport of water and small polar solutes, namely glycerol and urea [66].”

4. Figure 1: This should be improved. For example, Metal-based complex is not found in the figure.

Answer: The symbol of the metal-based compound was included in the figure.

5. Lines 388-388: The sentence is strange to me in grammar. 

Answer: The sentence “Being the most extensively studied and successful lipid-based nanosystem, a variety of liposomal formulations is already in biomedical use (Table 2) or under clinical trials “ was changed to “Liposomes as drug delivery systems have been extensively studied and, based on their ability to both incorporate hydrophilic or hydrophobic molecules, preclinical studies demonstrated an improved therapeutic performance of associated compounds. Many successful applications are already in clinical use (Table 2) or under clinical trials “.

6. Figure 9: The structure of 4-S-CAP is not correct (one more methylene group is needed). 

Answer: Thank you for your observation. The structure of 4-S-CAP was corrected.